# A Third Dose COVID-19 Vaccination in Allogeneic Hematopoietic Stem Cell Transplantation Patients

**DOI:** 10.3390/vaccines10111830

**Published:** 2022-10-29

**Authors:** Marika Watanabe, Kimikazu Yakushijin, Yohei Funakoshi, Goh Ohji, Hiroya Ichikawa, Hironori Sakai, Wataru Hojo, Miki Saeki, Yuri Hirakawa, Sakuya Matsumoto, Rina Sakai, Shigeki Nagao, Akihito Kitao, Yoshiharu Miyata, Taiji Koyama, Yasuyuki Saito, Shinichiro Kawamoto, Katsuya Yamamoto, Mitsuhiro Ito, Tohru Murayama, Hiroshi Matsuoka, Hironobu Minami

**Affiliations:** 1Division of Medical Oncology/Hematology, Department of Medicine, Kobe University Hospital Graduate School of Medicine, Kobe 650-0017, Japan; 2Division of Infectious Disease Therapeutics, Department of Microbiology and Infectious Diseases, Kobe University Graduate School of Medicine, Kobe 650-0017, Japan; 3R&D, Cellspect Co., Ltd., Morioka 020-0857, Japan; 4BioResource Center, Kobe University Hospital, Kobe 650-0047, Japan; 5Division of Molecular and Cellular Signaling, Kobe University Graduate School of Medicine, Kobe 650-0017, Japan; 6Laboratory of Hematology, Division of Medical Biophysics, Kobe University Graduate School of Health Sciences, Kobe 654-0142, Japan; 7Department of Hematology, Hyogo Cancer Center, Akashi 673-0021, Japan; 8Cancer Center, Kobe University Hospital, Kobe 650-0017, Japan

**Keywords:** SARS-CoV-2 vaccination, allogeneic hematopoietic stem cell transplantation, BNT162b2, mRNA-1273, COVID-19, vaccination, booster shot

## Abstract

We previously reported that a second dose of BNT162b2 was safe and effective for allogeneic hematopoietic stem cell transplantation (HSCT) patients. Here, we investigated the safety and efficacy of a third dose of COVID-19 mRNA vaccine in allogeneic HSCT patients. Antibody titers against the S1 spike protein were measured using the QuaResearch COVID-19 Human IgM IgG ELISA kit. The previous study included 25 allogeneic HSCT patients who received two doses of BNT162b2. Following the exclusion of three patients because of the development of COVID-19 (*n* = 2) and loss to follow-up (*n* = 1), the study evaluated 22 allogeneic HSCT patients who received a third dose of COVID-19 mRNA vaccine (BNT162b2 [*n* = 15] and mRNA-1273 [*n* = 7]). Median age at the time of the first vaccination was 56 (range, 23–71) years. Five patients were receiving immunosuppressants at the third vaccination, namely calcineurin inhibitors (CI) alone (*n* = 1), steroids alone (*n* = 2), or CI combined with steroids (*n* = 2). Twenty-one patients (95%) seroconverted after the third dose. None of our patients had serious adverse events, new-onset graft-versus-host disease (GVHD), or GVHD exacerbation after vaccination. A third dose of the BNT162b2 and mRNA-1273 COVID-19 vaccines was safe and effective for allogeneic HSCT patients.

## 1. Introduction

Severe acute respiratory syndrome coronavirus-2 (SARS-CoV-2) infection, leading to coronavirus disease 2019 (COVID-19), has been associated with high mortality among patients with hematological disorders [1]. The widespread use of COVID-19 mRNA vaccines has reduced mortality and morbidity in these patients [2]. We previously showed that a second dose of COVID-19 mRNA vaccines was safe and effective for Japanese hematopoietic stem cell transplantation (HSCT) patients [3]. but that approximately 25% of patients did not achieve seroconversion. Several reports revealed that some immunocompromised hosts, such as patients taking immunosuppressive agents or with hematological disease, had difficulty obtaining immunogenicity and that antibody titers against the S1 spike protein decreased early [4,5]. A recent systematic review and meta-analysis has shown that immune response to COVID-19 vaccines is impaired in HSCT patients, with a seropositive proportion after the second dose of about 79% [6]. Although a booster shot (third dose of vaccine) is widely accepted, research into the booster shot in HSCT patients remains unsatisfactory.

Here, we investigated the safety and efficacy of a third dose of COVID-19 mRNA vaccine in Japanese allogeneic HSCT patients.

## 2. Materials and Methods

### 2.1. Study Design

We prospectively evaluated the safety and efficacy of a third dose of COVID-19 mRNA vaccine in Japanese allogeneic HSCT patients who were enrolled in our previous study of a second dose of COVID-19 vaccine (BNT162b2) from March 2021 to August 2021 [3]. All patients had undergone allogeneic HSCT at Kobe University Hospital. They received a third dose of mRNA SARS-CoV-2 vaccine (BNT162b2 [Pfizer-BioNTech] or mRNA-1273 [Moderna] at the patient’s discretion). Peripheral blood samples were collected 1 to 4 weeks after the third vaccination, and before the third vaccination when possible. Patients who had previously developed COVID-19 confirmed by PCR positivity were excluded. We also evaluated antibody titers against nucleocapsid protein to exclude patients with asymptomatic COVID-19. Vaccine-related adverse events were graded according to Common Terminology Criteria for Adverse Events version 5.0, except for fever, which was categorized as follows: grade 1, 37.5–37.9 °C; grade 2, 38.0–38.9 °C; grade 3, 39.0–39.9 °C; and grade 4, >40.0 °C in the axilla. All patients measured their body temperature and documented their symptoms by themselves daily for at least one week, and thereafter adverse events were checked at their regular visits. The optimal optical density (O.D.) of anti-S1 IgG cut-off was 0.26, as previously reported, Ref. [7] and the O.D. of anti-nucleocapsid protein IgG cut-off was 0.7. This study was approved by the Kobe University Hospital Ethics Committee (No. B2056714, 1481) and conducted in accordance with the Declaration of Helsinki. All participants provided written informed consent for this study.

### 2.2. Measurement of Antibody Titers

A method for the precise measurement of antibody titers has been previously reported [3]. Briefly, after centrifugation of blood samples, serum antibody titers against S1 protein and nucleocapsid protein were measured using the QuaResearch COVID-19 Human IgM IgG ELISA kit (Cellspect, Inc., RCOEL961S1, Iwate, Japan) and QuaResearch COVID-19 Human IgM IgG ELISA kit (Nucleocapsid protein) (Cellspect, Inc., RCOEL961-N, Iwate, Japan), respectively, according to the manufacturer’s protocols.

### 2.3. Statistical Analysis

Antibody titers against S1 protein were not normally distributed. Accordingly, comparisons between the second and third doses were done using nonparametric methods (Wilcoxon signed ranks test). Antibody titers in each group were evaluated using the Mann–Whitney U test. Serum IgG and lymphocyte counts at the first and third vaccines were evaluated using the Wilcoxon signed rank test. All statistical tests were two-sided and were performed using GraphPad Prism software (version 9, San Diego, CA, USA) and EZR (Saitama Medical Center, Jichi Medical University, Saitama, Japan), a graphical user interface for R (The R Foundation for Statistical Computing, Vienna, Austria) [8], with *p* < 0.05 as the level of significance.

## 3. Results

### 3.1. Patient Characteristics

Our previous study included 25 allogeneic HSCT patients who received two doses of COVID-19 mRNA vaccine. For the present study, three patients were excluded because of the development of COVID-19 (*n* = 2) and loss to follow-up (*n* = 1), and 22 HSCT patients who received a third dose of COVID-19 mRNA vaccine (BNT162b2 [n = 15] and mRNA-1273 [n = 7]) were evaluated.

At the time this study was conducted, the third vaccine was given at least six months after the second vaccination in Japan. Seven patients were receiving immunosuppressants at the first vaccination, namely calcineurin inhibitors (CI) alone (*n* = 2, both tacrolimus), steroids alone (*n* = 3), or CI combined with steroids (*n* = 2, tacrolimus and cyclosporine). Two patients subsequently discontinued immunosuppressive drugs (one taking steroid alone and the second taking tacrolimus alone) before the third dose. During the course of this study, no new patients started immunosuppressive therapy. Consequently, five patients had taken immunosuppressants at the third vaccination. Patient characteristics at the third vaccination are shown in Table 1.

### 3.2. Efficacy

Median O.D. of S1 IgG titers after the second dose, and before and after the third dose were 0.540 (range, 0.016–1.991), 0.099 (range, 0.001–0.713) and 1.315 (range, 0.006–1.730), respectively (Figure 1). Although antibody titers were measured in only 16 patients before the third vaccination, 10 of these 16 patients (63%) had lost seroconversion. All 10 patients had significantly lower antibody titers after the second dose than the 6 patients who retained seroconversion (*p* = 0.0018). However, 21 patients (95%) underwent seroconversion after the third dose, and antibody titers after this dose were significantly higher than after the second (*p* = 0.0001).

In our previous study, patients taking immunosuppressants or steroids, with low IgG or with low lymphocyte counts, had low S1 antibody titers. In present study, however, six of seven patients treated with steroids or CI also seroconverted after the vaccination. Two patients discontinued immunosuppressants; one of these two patients who discontinued steroid after the second vaccine dose obtained seroconversion for the first time after the third dose. The second patient, who discontinued tacrolimus after the second vaccine dose, had already developed immunogenicity after the second dose. Of 22 patients, there were five patients with lymphocytes <1000/µL and only two patients with serum IgG < 600 mg/dL, although lymphocyte counts and serum IgG did not significantly differ between the first and third vaccine (*p* = 0.949 and 0.673, respectively). One patient who could not achieve seroconversion even after the third vaccination had received steroid alone, and had a serum IgG level of 173 mg/dL. Antibody titers after the third vaccine did not significantly differ between vaccine types and with or without immunosuppressants (*p* = 0.58 and 0.368, respectively) (Figure 2); nevertheless, some patients with immunosuppressants had low antibody titers, in contrast with patients without immunosuppressants, all but one of whom had sufficient antibody titers.

We also evaluated antibody titers against nucleocapsid protein to investigate previous undiagnosed COVID-19. All patients analyzed in this study had lower anti-nucleocapsid IgG than the threshold for positivity (<0.7). The median O.D. was 0.061 (range, 0.014–0.41).

### 3.3. Safety

Adverse events profile is shown in Table 2. In our previous study, all adverse events were mild (grade 1 only), while in this study three patients had grade 2 fever (38–38.9 °C), although no patient had ≥grade 3 or other serious adverse events. All adverse events resolved within several days. There was no new-onset graft-versus-host disease (GVHD) or GVHD exacerbation after vaccination. The most frequent adverse event was mild (grade 1) pain at the injection site. One patient developed transient mild skin rash 17 days after the third vaccination, which resolved in a few days before skin biopsy.

## 4. Discussion

We evaluated the efficacy and safety of a third dose of vaccination in Japanese allogeneic HSCT patients who had received two doses of BNT162b2 vaccine. Three important results were obtained: first, 95% of these HSCT patients achieved seroconversion, among whom were patients taking immunosuppressants or steroids; second, although some patients had lost seroconversion before the third does, antibody titers after the third dose were much higher than after the second dose for all patients except one; and finally, all adverse events were mild.

In our previous study, we reported that allogeneic HSCT patients had a range of antibody titers after a second dose and that 19/25 (76%) patients obtained seroconversion. Some patients had an extremely low antibody titer, which was associated with low serum IgG, low lymphocyte count, or with or without immunosuppressants and steroids [3]. Several studies reported seroconversion rates of 75–86% in allogeneic HSCT patients [5,9,10,11], and a recent meta-analysis study also revealed a diminished response in HSCT patients [6]. Regarding the third vaccination, Kimura et al. reported that 89% of allogeneic HSCT patients obtained seroconversion after the third dose of vaccine [12]. They pointed out that chronic kidney disease, haploidentical donor status, and median lymphocyte counts at the third dose were significantly associated with suboptimal antibody response. In our study, 21/22 (95%) patients achieved seroconversion after a third dose of vaccination. Our patient number was too small to evaluate factors related to a poor response to vaccination.

It is known that COVID-19 antibody titer decreases gradually in elderly people and patients with complications [13], and our data indicated that over a half of HSCT patients who could be evaluated for immunogenicity before the third vaccine had lost immunogenicity approximately 200 days after the second vaccine. Lower antibody titer after the second dose is apparently associated with short-term immunogenicity. In the present study, we excluded two patients who had developed COVID-19. Both had obtained seroconversion after the second dose (antibody titers were 0.384 and 0.540, respectively), and were infected approximately 6 months after the second dose, which implies that they had lost seroconversion over time after the second vaccination and then contracted COVID-19. Two patients had mild symptoms—one was treated with sotrovimab and the second received symptomatic medication only—and both recovered without exacerbation.

We observed that five of six HSCT patients who had not obtained immunogenicity after the second dose of BNT162b2 mRNA obtained seroconversion after the third SARS-CoV-2 vaccination. A previous study pointed out that immunosuppressive drug use and lymphocyte count were associated with low antibody titers [14]; in contrast, however, our data did not indicate significant relationships. These results may be associated with the small number of patients treated with immunosuppressants, two of whom were able to discontinue immunosuppressive drugs. Redjoul et al. found that about a half of HSCT patients could not develop immunogenicity even after a third vaccination [15], which may suggest that B-cell immune status at vaccination is an important factor for COVID-19 immunogenicity. Although in our data serum IgG and lymphocyte count between the first and third vaccine did not significantly differ, all patients except one achieved seroconversion. Maillard et al. reported that antibody levels following the third dose significantly increased in HSCT patients with a detectable but weak response before the third dose, and that 41% of HSCT patients with no detectable response to the second dose had some response to the third dose, Ref. [16] indicating that repeated vaccination might bring immunogenicity to HSCT patients with poor humoral immunity.

As in our previous study, there were no severe adverse events in this study (≥grade 3). The profile of adverse event profile was slightly different compared to our previous study, which might be attributed to differences in vaccine types. Kimura et al. mentioned a 5.4% incidence of GVHD induction or worsening after the third vaccine, Ref. [12] which we did not experience in our cohort. This difference might be associated with our exclusion of acute and chronic GVHD patients, ethnicity and number of patients. Indeed, one of our patients developed a skin rash which resolved before skin biopsy for GVHD diagnosis. The relationship between this skin rash and the third vaccine is unknown. Careful observation over a longer period is required.

This study had several limitations. First, the number of patients was small. Second, we were unable to determine safety for patients with acute GVHD or active chronic GVHD. Third, the precise cut-off value of antibody titer remains unclear and we were unable to determine the meaning of the decline in immunogenicity. Fourth, the antibody titers were measured in a single experimental session due to the limited volume of samples. However, we believe that the third vaccination is promising for patients who have undergone allogeneic HSCT.

## 5. Conclusions

We found that 95% of patients obtained higher immunogenicity with a third dose of SARS-CoV-2 vaccine than with the second dose, and with safety. Further research to support our present data is required.

## Figures and Tables

**Figure 1 vaccines-10-01830-f001:**
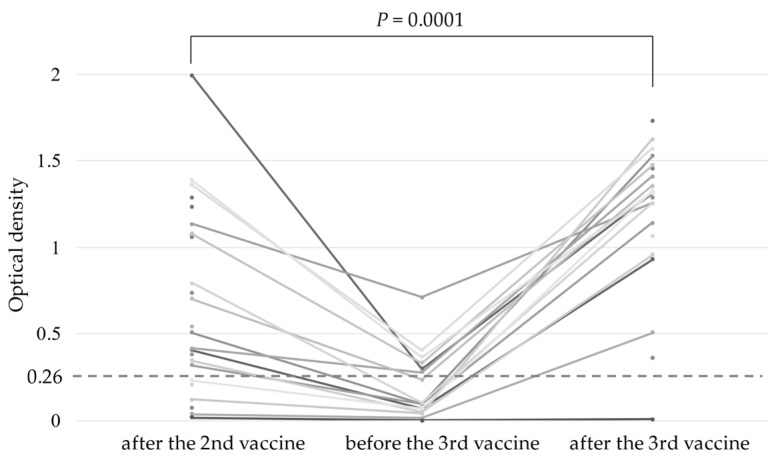
Anti-S1 antibody titers after vaccination. Anti-S1 antibody response after the second dose, prior to the third dose and 1–4 week(s) after the third dose in patients undergoing hematopoietic stem cell transplantation. The broken line indicates the cut-off value (0.26) for seroconversion.

**Figure 2 vaccines-10-01830-f002:**
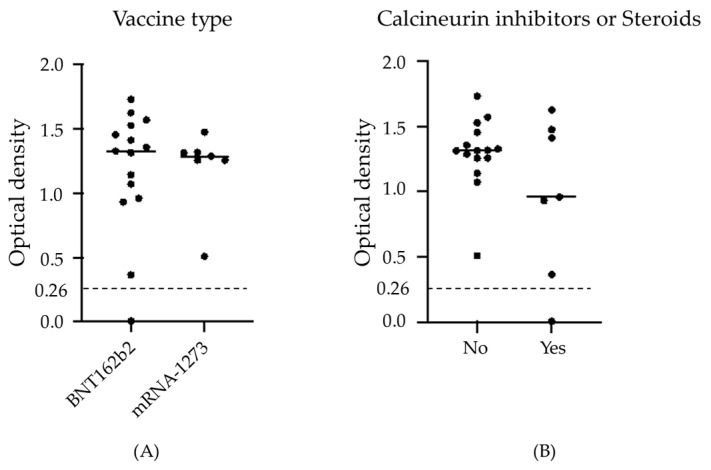
Comparison of antibody titers after the third vaccine in each group: (**A**) vaccine type, and (**B**) with or without calcineurin inhibitors or steroids. Antibody titers after the third vaccine did not significantly differ between vaccine types or with or without immunosuppressants. The broken line indicates the cut-off value (0.26) for seroconversion.

**Table 1 vaccines-10-01830-t001:** Patient characteristics at the third vaccination.

Median age at vaccination (range, years).	56 (23–71)	
Median IgG (range, mg/dL)	957 (173–2126)	
Median lymphocyte count (range, /µL)	1814 (32–4700)	
Median duration (range)		
Transplantation—third vaccine	1842 (378–4279)	
Second vaccine—third vaccine	219 (194–258)	
Sex	*n*	%
Female	10	45
Male	12	55
Conditioning		
MAC; myeloablative conditioning	6	27
RIC; reduced-intensity conditioning	16	73
Disease		
AML; acute myeloid leukemia	8	36
ALL; acute lymphoblastic leukemia	6	27
ML; malignant lymphoma	5	23
Others	3	14
Immunosuppressants		
Tacrolimus alone	1	5
Tacrolimus + steroid	1	5
Cyclosporine A + steroid	1	5
Steroid alone	2	9
No use	17	77

**Table 2 vaccines-10-01830-t002:** Adverse events.

Adverse Event	Grade 1	Grade 2	Total	%
Fever	1	3	4	18
Pain	21	0	21	95
Redness	4	0	4	18
Swelling	4	0	4	18
Headache	7	0	7	32
Fatigue	9	0	9	41
Chills	3	0	3	14
Muscle pain	2	0	2	9
Joint pain	0	0	0	0
Vomiting	0	0	0	0
Diarrhea	1	0	1	5

## Data Availability

The data presented in this study are available on request from the corresponding author.

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
