# Peer review of "A Third Dose COVID-19 Vaccination in Allogeneic Hematopoietic Stem Cell Transplantation Patients"

_vaccines, 2022, doi:10.3390/vaccines10111830_

Round 1

Reviewer 1 Report

Please describe the follow up in more detail as the information in the abstract does not match the remainder of the manuscript. Furthermore, the follow up in the abstract is listed as 7 days, which seems very short and again does not match the remainder of the manuscript. Please state at which time point after 3rd vaccination the assessment on side effects was done and what the follow up approach was (when, how often).

Author Response

Thank you for your important comment. Accordingly, we add the following sentence; All patients measured their body temperature and documented their symptoms by themselves daily for at least one week, and thereafter adverse events were checked at their regular visits.(p.2 L.79-81)

Reviewer 2 Report

The manuscript by Watanabe et al, reports the effect of the 3rd dose of COVID-19 vaccination on allogeneic hematopoietic stem cell transplantation patients. The manuscript is well-written, and the results are very clear. The conclusion is also well supported by the data.  I have no major concerns about the manuscript, except that the number of patients considered is quite small. Also, I want the authors to explain the reason behind the grade 3 fever of a few patients compared to their previous study. Overall, the manuscript can be accepted for publication after a minor revision.  

Author Response

Thank you for your insightful comment. As you say, the number of patients was too small. We speculate that the fact of a little different profile of adverse events compared to our previous study might be attributed to differences in vaccine types. (BNT162b2 and mRNA-1273) Accordingly, we added the sentence in the discussion part; “The profile of adverse event profile was slightly different compared to our previous study, which might be attributed to differences in vaccine types.” (p.6 L.218-219)